# Long-Term Effects of Semaglutide and Sitagliptin on Circulating IGFBP-1, IGFBP-3 and IGFBP-rp1: Results from a One-Year Study in Type 2 Diabetes

**DOI:** 10.3390/ijms262110404

**Published:** 2025-10-26

**Authors:** Eszter Dániel, Ferenc Sztanek, Sára Csiha, Balázs Ratku, Sándor Somodi, György Paragh, Mariann Harangi, Hajnalka Lőrincz

**Affiliations:** 1Division of Metabolism, Department of Internal Medicine, Faculty of Medicine, University of Debrecen, H-4032 Debrecen, Hungary; 2Doctoral School of Health Sciences, University of Debrecen, H-4032 Debrecen, Hungary; 3Department of Clinical Basics, Faculty of Pharmacy, University of Debrecen, H-4032 Debrecen, Hungary; 4Department of Emergency Medicine, Faculty of Medicine, University of Debrecen, H-4032 Debrecen, Hungary; 5Institute of Health Studies, Faculty of Health Sciences, University of Debrecen, H-4032 Debrecen, Hungary

**Keywords:** semaglutide, sitagliptin, insulin-like growth factor-binding protein, insulin, C-peptide, waist circumference, lipid parameters, C-reactive protein, type 2 diabetes mellitus

## Abstract

The role of insulin-like growth factor-binding proteins (IGFBPs) in the regulation of carbohydrate metabolism and the development of complications is well established; however, the impact of the glucagon-like peptide-1 receptor agonist semaglutide on IGFBPs has not been previously investigated. We aimed to examine the effects of semaglutide and dipeptidyl peptidase-4 inhibitor sitagliptin therapy on serum levels of IGFBP-1, IGFBP-3, and IGFBP-rp1, and to analyze their associations with anthropometric variables and markers of carbohydrate and lipid metabolism. In this prospective study, we enrolled 34 patients with type 2 diabetes mellitus (T2DM) on metformin monotherapy and 31 age-, sex- and BMI-matched controls. Among the patients, 18 received semaglutide, and 16 were treated with sitagliptin. Anthropometric and laboratory assessments were performed at baseline, 26 and 52 weeks. IGFBP levels were measured using ELISA. Both semaglutide and sitagliptin treatment significantly increased IGFBP-1 levels. IGFBP-3 levels were significantly decreased following sitagliptin therapy. No significant change in IGFBP-rp1 levels was observed with either treatment. Based on multiple regression analysis, the best predictors of IGFBP-1 were insulin and hsCRP, while the best predictor of IGFBP-3 was LDL-C level. Our findings suggest that semaglutide and sitagliptin may exert favorable effects on the GH/IGF-1 axis, potentially contributing to their beneficial metabolic outcomes in patients with T2DM.

## 1. Introduction

The prevalence of type 2 diabetes mellitus (T2DM) continues to rise and represents a major risk factor for cardiovascular diseases, which remain the leading cause of mortality worldwide [1]. Dipeptidyl peptidase-4 inhibitors (DPP-4is) and glucagon-like peptide-1 receptor agonists (GLP-1 RAs) are widely used in pharmacological therapy due to their action on the incretin pathway. Both drug classes possess beneficial effects beyond glycemic control, contributing to the reduction of both microvascular and macrovascular complications [2,3]. The GLP-1 RA semaglutide is well known for its anti-inflammatory properties and its beneficial effects on lipid metabolism [4]. Furthermore, the SUSTAIN-6 and PIONEER 6 trials have confirmed its beneficial effects on cardiovascular risk in patients with T2DM [5,6].

The growth hormone/insulin-like growth factor 1 (GH/IGF-1) axis plays a key role in growth and development, as well as in the maintenance of physiological glucose homeostasis, as demonstrated by numerous previous studies [7]. IGF-binding proteins (IGFBPs), of which six have been identified to date—with the seventh referred to as IGFBP-related protein 1 (IGFBP-rp1)—are responsible for binding insulin-like growth factor 1 (IGF-1), thereby facilitating its transport, prolonging its half-life, and regulating its degradation, organ-specific distribution, and biological activity. In addition to these functions, IGFBPs also exert IGF-1-independent effects [8]. Serum IGFBP-1 levels are positively correlated with insulin sensitivity, while negative correlations have been reported with body mass index (BMI), waist-to-hip ratio, and fasting insulin levels [9,10]. Previous studies have reported altered IGFBP-1 levels in patients with type 2 diabetes, with several studies describing increased concentrations, while others have demonstrated decreased levels depending on patient characteristics [11]. IGFBP-3 is the primary carrier protein for IGF-1 and exerts deleterious effects on insulin resistance and adipose tissue function by promoting the production of pro-inflammatory cytokines in adipocytes [12,13,14]. IGFBP-rp1 level positively correlates with the degree of insulin resistance [15], and its concentration is elevated in newly diagnosed T2DM patients [16].

Data regarding the effects of antidiabetic agents on IGFBP levels in T2DM are currently very limited. Treatment with sulfonylurea or combined sulfonylurea plus insulin significantly reduced IGFBP-1 levels, whereas once-daily insulin therapy appeared to have no effect on IGFBP-1 concentrations [17]. In contrast, multiple daily insulin injections were associated with increased IGFBP-1 levels [18]. According to earlier findings, metformin treatment increases circulating IGFBP-2 levels [19]. A previous Japanese study reported that DPP-4is reduce IGFBP-3 concentrations [20]. Empagliflozin therapy has been shown to significantly increase IGFBP-1 levels in T2DM patients [21]. Analysis of the CREDENCE trial data indicated that three years of canagliflozin treatment did not affect IGFBP-3 concentrations [22]. Tirzepatide therapy increased both IGFBP-1 and IGFBP-2 levels in patients with T2DM, while the comparator drug, the GLP-1 RA dulaglutide, did not influence the levels of the studied IGFBPs [23]. In summary, these results highlight the need for further clinical studies to clarify the disparate effects of antidiabetic drugs on the IGFBP system.

As the effect of semaglutide on serum IGFBPs has not yet been investigated, we aimed to examine the levels of IGFBP-1, IGFBP-3, and IGFBP-rp1 in T2DM patients receiving metformin monotherapy, as well as following one year of semaglutide or sitagliptin treatment. In addition, we aimed to explore the associations between serum levels of the studied IGFBPs and anthropometric parameters, as well as markers of carbohydrate and lipid metabolism. We hypothesized that both drugs would increase IGFBP-1 levels and decrease IGFBP-3 and IGFBP-rp1 concentrations.

## 2. Results

### 2.1. Baseline Anthropometric and Routine Laboratory Parameters of the Enrolled T2DM Patients and Non-Diabetic Controls

There were no significant differences in age, gender, BMI and waist circumference between the three groups (Table 1). Fasting blood glucose, fructosamine, hemoglobin A1c (Hb_A1c_), and C-peptide levels were significantly higher in both T2DM groups compared to controls (all *p* < 0.05). Baseline insulin level was higher in the sitagliptin group compared to controls (all *p* < 0.05). Baseline hsCRP levels were lower in the semaglutide group than in the non-diabetic controls (*p* < 0.05). There was no difference observed between the baseline data of T2DM patients on semaglutide and sitagliptin treatments.

### 2.2. Changes in Anthropometric and Routine Laboratory Parameters of T2DM Patients After a 52-Week Semaglutide Treatment

Changes in the anthropometric and laboratory parameters of enrolled patients receiving semaglutide treatment are summarized in Table 2. A marked reduction was detected in BMI, waist circumference, fasting glucose, fructosamine, Hb_A1c_, and low-density lipoprotein cholesterol (LDL-C) at week 26 and week 52 compared to baseline, respectively (all *p* < 0.05). Non-high-density lipoprotein cholesterol (non-HDL-C) also reduced at week 52 compared to baseline (*p* < 0.05). HDL-C levels increased at week 26 and remained elevated at week 52 (all *p* < 0.05). Insulin, C-peptide, high-sensitivity C-reactive protein (hsCRP), triglyceride, total cholesterol, sICAM-1, sVCAM-1, and renal and liver function parameters remained unchanged during the follow-up.

### 2.3. Changes in Anthropometric and Routine Laboratory Parameters of T2DM Patients After 52-Week Sitagliptin Treatment

Changes in anthropometric and laboratory parameters of enrolled patients receiving sitagliptin treatment for 52 weeks are summarized in Table 3. A slight decrease in BMI, fasting glucose, fructosamine, Hb_A1c_, and Apolipoprotein AI (ApoAI) was observed during the follow up (all *p* < 0.05). There were no significant changes in insulin, C-peptide, creatinine, estimated glomerular filtration rate (eGFR), hsCRP, liver enzymes, lipid parameters, and sICAM-1 and sVCAM-1 levels at week 52.

### 2.4. Baseline Concentrations and Changes in IGFBP-1, IGFBP-3 and IGFBP-rp1 in Patients with T2DM After 52-Week Semaglutide and Sitagliptin Treatment

Baseline IGFBP-1 levels were significantly lower in both treatment groups compared to controls (semaglutide: 3072.2 (1041.5–8827.6) vs. controls: 8392.3 (4382.7–16,505.0) pg/mL, *p* = 0.031; and sitagliptin: 4187.5 (466.7–14,992.9) vs. controls: 8392.3 (4382.7–16,505.0) pg/mL, *p* = 0.044; respectively) (Figure 1a). Serum IGFBP-1 significantly increased during the 52-week follow-up in the semaglutide group (from 3072.2 (1041.5–8827.6) pg/mL to 6811.3 (2009.8–25,135.2) pg/mL; *p* = 0.038). Similarly, IGFBP-1 was also increased in the sitagliptin arm and was comparable to the controls’ concentration (from 4187.5 (466.7–14,992.9) pg/mL to 8958.9 (2560.8–39,399.2) pg/mL; *p* = 0.023) (Figure 1a).

Serum IGFBP-3 tended to be lower after semaglutide treatment but did not reach significance (from 1130.8 ± 282.3 ng/mL to 1067.7 ± 294.9 ng/mL, *p* = 0.155). In the sitagliptin group, there was a significant decrease during the follow-up (from 1271.4 ± 342.5 ng/mL to 987.3 ± 294.9 ng/mL, *p* = 0.008). Baseline IGFBP-3 levels were comparable between both treatment groups and controls (semaglutide: 1130.8 ± 282.3 vs. sitagliptin: 1271.4 ± 342.5 vs. controls: 1107.9 ± 216.1 ng/mL, *p* > 0.050) (Figure 1b).

Serum IGFBP-rp1 remained unchanged during semaglutide treatment during the one-year follow-up (from 149.5 ± 31.4 ng/mL to 145.6 ± 34.7 ng/mL, *p* = 0.679). Sitagliptin treatment reduced serum IGFBP-rp1 during the follow-up but did not reach the significance (from 163.2 ± 51.0 ng/mL to 138.6 ± 24.8 ng/mL, *p* = 0.089). Baseline IGFBP-rp1 levels were similar between both treatment groups and controls (semaglutide: 149.5 ± 31.4 vs. sitagliptin: 163.2 ± 51.0 vs. controls: 158.1 ± 25.3 ng/mL, *p* > 0.050) (Figure 1c).

### 2.5. Correlations of Baseline IGFBP-1, IGFBP-3 and IGFBP-rp1 in T2DM Patients and Controls

Baseline serum IGFBP-1 correlated negatively with waist circumference (*p* = 0.008), C-peptide (*p* = 0.009), insulin (*p* = 0.002), creatinine (*p* = 0.038) and triglyceride (*p* = 0.05), while it correlated positively with hsCRP (*p* = 0.005) and ApoAI (*p* = 0.017) in the overall study population (Figure 2a). Baseline serum IGFBP-3 correlated positively with lipid parameters including cholesterol (*p* = 0.002), non-HDL-C (*p* = 0.048), LDL-C (*p* = 0.002) and Apolipoprotein B100 (ApoB100) (*p* < 0.001) across the whole study population. In addition, there was a positive correlation between baseline soluble intercellular adhesion molecule-1 (sICAM-1) and IGFBP-3 (*p* = 0.042) (Figure 2b) in the whole study population. Moreover, baseline IGFBP-rp1 correlated positively with BMI (*p* = 0.041), C-peptide (*p* = 0.043), and sICAM-1 (*p* = 0.050) overall (Figure 2c). Correlations of baseline IGFBP-1, IGFBP-3 and IGFBP-rp1 with anthropometric and laboratory parameters in the diabetic and control groups separately are summarized in Appendix A, respectively.

### 2.6. Multiple Regression Analyses

To determine the significant predictor(s) of IGFBP-1 and IGFBP-3, backward stepwise multiple regression analyses were performed. Two different models were designed based on data of univariate analyses, as indicated in Figure 2. Model A (IGFBP-1) included waist circumference, creatinine, hsCRP, insulin, C-peptide, triglyceride and ApoAI in the dependent variable list in overall subjects. Insulin (β_standardized_ = −0.620; *p* = 0.002) and hsCRP (β_standardized_ = 0.424; *p* < 0.001) were identified as predictors of IGFBP-1. In Model B (IGFBP-3) cholesterol, nonHDL-C, LDL-C, ApoB100, and sICAM-1 were incorporated into the dependent variable list in overall subjects. LDL-C was found to be the best predictor of IGFBP-3 (β_standardized_ = 0.387; *p* = 0.016). None of the studied parameters were found to be a significant predictor of IGFBP-rp1.

## 3. Discussion

To our knowledge, this is the first clinical study to evaluate the effect of one-year semaglutide and sitagliptin treatment on serum IGFBP-1, IGFBP-3 and IGFBP-rp1 levels in overweight and obese patients with T2DM receiving metformin monotherapy. Our findings demonstrate the IGFBP-1-increasing effect of long-term semaglutide and sitagliptin therapy. Although we expected comparable effect of semaglutide and sitagliptin on IGFBP-3 levels, only sitagliptin treatment could reduce its level significantly. The lack of changes in IGFBP-rp1 levels after semaglutide and sitagliptin treatment are also a novel result. Overall, our results highlight the beneficial effects of long-term semaglutide and sitagliptin treatment on the GH/IGF-1 axis.

IGFBP-1 is the first member of the IGFBP superfamily with multiple endocrine roles in metabolism through IGF-1-dependent and -independent actions [17]. IGFBP-1 is primarily secreted in the liver where the synthesis is stimulated by multiple factors like glucagon, oestrogen, glucocorticoids, and cytokine and inhibited by insulin [24]. Therefore, serum IGFBP-1 represents a composite biomarker modulated by the suppressive effect of hepatic portal insulin, insulin sensitivity, as well as the effects of varied stimulatory mediators [17]. In our study, baseline IGFBP-1 was significantly lower in patients with T2DM compared to controls, which is partly consistent with previous publications. In these papers, low serum IGFBP-1 concentration was associated with the components of metabolic syndrome [11] and predicted the development of diabetes [11,25,26]; however, other studies did not support these data [27]. Indeed, high IGFBP-1 predicted diabetes onset in a high-risk prediabetic cohort over two years compared to healthy population [27]. Additionally, an observation on zebrafish islets showed high levels of recombinant IGFBP1 increased β-cell regeneration by promoting α- to β-cell transdifferentiation, which may serve to explain the contradictory results [28]. However, extrapolating the results to other species is challenging due to interspecies variability.

To date, the long-term effects of antidiabetic drugs, especially GLP-1 RAs, on circulating IGFBPs have been incompletely studied in T2DM. Dual glucose-dependent insulinotropic polypeptide and GLP-1 RA tirzepatide therapy increased both serum IGFBP-1 and IGFBP-2 in diabetic patients, while the comparator drug, the GLP-1 RA dulaglutide, did not influence the levels of the IGFBPs studied at 26 weeks [23]. In our study, serum IGFBP-1 increased at 26 weeks and remained high at the one-year check-up both in semaglutide and sitagliptin treatments. The results may highlight the possible differences in the effect on IGFBPs among GLP-1 RAs. Our results may verify the improvement of the insulin-sensitizing effect of IGFBP-1 in T2DM. Parallel with these results, the markers of carbohydrate metabolism, i.e., insulin and C-peptide, were significantly improved; moreover, BMI and waist circumference markedly reduced during the one-year semaglutide treatment. In univariate analyses, baseline IGFBP-1 showed correlations with waist circumference and glycemic parameters in overall subjects. According to the multiple regression analysis baseline insulin and hsCRP were the best predictors of baseline serum IGFBP-1. These data are corroborated by previous observations [11,26,29].

IGFBP-3 is the most abundant protein among IGFBPs in the bloodstream, with significant bioactivity in both an IGF-1-dependent and -independent manner [7]. IGFBP-3 forms a ternary complex with IGF-1 and the acyl label subunit promoting the binding of IGF-1 its cell surface receptors and activation of its associated downstream signaling cascade [30]. Due to IGF-1-independent action, IGFBP-3 has been associated with impaired glucose tolerance and insulin resistance in mouse models and in vitro studies as well [14,31]. Additionally, IGFBP-3 inhibited the phosphorylation of the insulin receptor in adipocytes, thereby affecting insulin-stimulated glucose uptake [13]. These findings suggest that IGFBP-3 might exert direct insulin-antagonistic effects, potentially elevating the incidence of diabetes, though independently of its biological impact on IGF-1. Several clinical studies have confirmed this data, and elevated levels of IGFBP-3 have been proven in diabetic patients. In these studies, high circulating concentration of IGFBP-3 strongly correlated with the incidence of T2DM in older women in the Cardiovascular Health Study [32] and in the Nurses’ Health Study [26]. Contradictory findings have also been published regarding IGFBP-3 levels in obesity, since higher IGFBP-3 was shown in obese individuals, but other papers did not verify these findings [33]. In our study, baseline serum IGFBP-3 was tendentiously but not significantly higher in T2DM patients compared with controls, possibly because a BMI-matched overweight cohort served as controls. Unexpectedly, semaglutide treatment did not change IGFBP-3 levels over time; only sitagliptin treatment reduced serum IGFBP-3 during the 52-week follow-up. DPP-4is enhance the levels of active forms of both GLP-1 and GIP, restoring the incretin balance [34], which leads to reduced IGFBP-3 levels in the circulation. On the contrary, GLP-1 RAs in supraphysiological doses provide GLP-1 signaling only [35]. The improvement in glucose metabolism markers is not accompanied by a decrease in IGFBP-3 levels, suggesting that the beneficial effect of semaglutide is presumably not mediated through a reduction in IGFBP-3. Both pharmacological therapies effectively reduced BMI and normalized glucose parameters in diabetic patients over the one-year follow-up period. Through mechanisms likely independent of IGF-1, IGFBPs appear to be strongly associated with glucose homeostasis in diabetes. Therefore, the observed changes in IGFBP levels may reflect secondary effects of improved metabolic control rather than direct drug-induced modulation via unidentified pathways.

In vitro and clinical studies reported the potential regulatory role of IGFBP-3 on the lipid metabolism. IGFBP-3 inhibited adipocyte differentiation in a PPARγ-dependent manner, and these processes were related to decreased lipid accumulation in mature adipocytes [36]. Moreover, IGFBP-3 suppressed the Smad signaling pathway in 3T3-L1 adipocytes, and IGFBP-3 influenced adipogenic differentiation within the cells through the regulatory effect of transforming growth factor-β [37]. A large population-based German study investigated almost 3000 subjects, and it was demonstrated that IGFBP-3 positively related to total cholesterol, LDL-C and triglyceride levels, whereas no longitudinal relationship was found [38]. Also, Kawachi et al. found that total cholesterol, triglyceride, visceral adipose tissue, and carotid intima media thickness were positively correlated with IGFBP-3, whereas no correlation was described between HDL-C and IGFBP-3 in healthy Japanese men [39]. These findings are in line with our results, since baseline serum IGFBP-3 strongly correlated with ApoB100-containing lipid parameters but not with HDL-C and ApoAI. Multiple regression analysis showed that LDL-C was the independent predictor of IGFBP-3 in overall participants. Our results suggest that elevated levels of IGFBP-3 are linked to diabetic dyslipidemia. Therefore, measurement of serum IGFBP-3 may serve as a potential surrogate biomarker in cardiovascular risk prediction and may open up potential opportunities for personalized medicine in diabetic patients. Further large-scale studies are needed to clarify these findings.

IGFBP-pr1 is a secretory glycoprotein from different tissues which shares less structural homology with IGFBP-1 to -6 and exhibits modest affinity to IGF-1 [40]. Previous studies have demonstrated that higher serum IGFBP-rp1 is involved in the early defect of insulin actions; therefore, it serves as a marker of insulin resistance, metabolic syndrome, T2DM, and related cardiovascular complications [41]. However, inconsistent results have also been published. Gu et al. [15] demonstrated no statistically significant difference in serum IGFBP-rp1 levels among Swedish subjects with normal glucose tolerance and in patients with T2DM, which is in line with our results. In our study, serum concentrations of IGFBP-rp1 did not significantly change with either treatment indicating that long-term GLP-1 RA and DPP-4i treatment did not affect the level of this IGFBP. In addition, only marginally significant correlations were observed between baseline BMI, C-peptide, sICAM-1 and IGFBP-rp1, which suggests that serum IGFBP-rp1 may not play a pivotal role in glycemic regulation in this study population.

Despite the appropriate statistical power, the relatively small sample size could be one of the limitations of this study. Enrollment of larger patient population and a multi-center study design may improve the power of the study. The substantial variability observed in serum IGFBP concentrations is likely attributable to the limited sample size and the large inter-individual heterogeneity. While a 52-week follow-up period is considered relatively long in a clinical trial, the effects of long-term changes in serum IGFBP levels on cardiovascular outcomes could only be assessed in a study with a longer follow-up period. Lack of racial diversity and the narrow age range of the enrolled patients restrict the generalizability of our results. Another limitation of the study may be that lifestyle factors—including diet, physical activity, and potentially other habits such as sleep patterns—were not strictly controlled. Since these factors can independently exert significant effects on glucose metabolism, insulin sensitivity, and other metabolic parameters, it cannot be excluded that some of the observed results were influenced by these variables. Nevertheless, the randomized study design, strict inclusion and exclusion criteria, and the long-term follow-up of the therapeutic regime are strengths of our investigation.

## 4. Materials and Methods

### 4.1. Patient Enrollment

A total of 34 overweight or obese patients with T2DM were enrolled in this single-center, 52-week, prospective study from the diabetes outpatient clinic at the Department of Internal Medicine, Faculty of Medicine, University of Debrecen, Hungary. The study was designed to evaluate the efficacy of the subcutaneous semaglutide, a GLP-1 receptor agonist, added to metformin monotherapy, compared with the combination of baseline therapy with the DPP-4i sitagliptin. Due to the small sample size, we used a simple randomization method [42,43] using the GraphPad online calculator for randomization (Dotmatics, La Jolla, CA, USA, https://www.graphpad.com/quickcalcs/randomize1/ (accessed on 12 May 2024)) and assigned 20 subjects to each of the two groups by a random sequence. Later, due to the distinct exclusion criteria, six patients were excluded from the study (Appendix A). After randomization, 18 patients (12 men, 6 women) started once-weekly subcutaneous semaglutide, and 16 patients (7 men, 9 women) started once-daily oral sitagliptin. In the semaglutide group, the target dose was reached by week 8 after a three-step titration period (0.25 mg/week for 4 weeks, then 0.5 mg/week, and 1.0 mg/week from week 8). Sitagliptin was administered at a fixed dose of 100 mg daily from baseline. Participants were instructed to maintain their usual dietary habits and levels of physical activity throughout the study period. No adverse events related to the two drugs were observed. All patients were carefully monitored by the physician, with both physical and psychological status assessed regularly. In the semaglutide-treated group, some patients experienced mild, transient gastrointestinal symptoms at the beginning of the treatment; however, these resolved spontaneously within a short time and did not require dose adjustment or discontinuation of therapy. Sitagliptin was also well tolerated, and no adverse events necessitating intervention were observed.

In addition, 31 age-, sex- and BMI-matched obese, non-diabetic subjects (10 men, 21 women) without medication served as controls and were recruited from the obesity outpatient clinic at the Department of Internal Medicine, Faculty of Medicine, University of Debrecen, Hungary. Overweight and obesity were defined according to the World Health Organization criteria, with overweight classified as a BMI between 25.0 and 29.9 kg/m^2^, and obesity as a BMI of 30.0 kg/m^2^ or higher. Non-diabetic status was confirmed using 75 g oral glucose tolerance test. In T2DM patients, venous blood samples were collected immediately before the start of treatment and at routine check-ups after 26 and 52 weeks; in the controls, samples were collected at presentation to the obesity outpatient clinic. A detailed study design flowchart of enrolled participants is depicted in Appendix A. All patients gave written informed consent to participate prior to enrollment. The study protocol was approved by the Regional Ethics Committee of the University of Debrecen (protocol code: RKEB/IKEB: 4739/2017, date of approval: 20 February 2017). The study was conducted in accordance with the Helsinki Declaration.

### 4.2. Inclusion and Exclusion Criteria

The main inclusion criteria were age over 45 years with at least a 5-year history of diabetes, inadequate glycaemic control with baseline metformin monotherapy (Hb_A1c_: 6.5–10%), and eGFR > 60 mL/min/1.73 m^2^ in T2DM patients. The main exclusion criteria were previous treatment with a GLP-1 RA or DPP-4i, current smoking, alcohol consumption, pregnancy, lactation, type 1 diabetes, acute pancreatitis, and active malignant disease (remission of previous cancer >5 years) including pancreatic or thyroid cancer. Further exclusion criteria were documented coronary or cerebrovascular events, clinically significant heart failure (NYHA stage II-IV), severe liver failure (Child Pugh stage B and C), chronic use of systemic glucocorticoids or immunosuppressive drugs, advanced diabetic retinopathy (proliferative stage) and diabetic nephropathy, or severe renal impairment due to other causes (eGFR < 60 mL/min/1.73 m^2^ and/or significant proteinuria).

### 4.3. Laboratory Analyses

Venous blood samples were taken after an overnight fast in the morning using Vacuette^®^ tubes (Greiner Bio-One, Mosonmagyaróvár, Hungary). Serum and plasma samples were prepared after 30 min rest by centrifugation. Routine laboratory parameters including triglycerides, total cholesterol, LDL-C, HDL-C, ApoAI, ApoB100, fructosamine, creatinine, glucose, insulin, C-peptide, Hb_A1c_, eGFR, hsCRP and liver enzymes were measured with standard laboratory techniques from fresh sera using a Cobas c501 autoanalyzer (Roche Ltd., Mannheim, Germany) according to the manufacturer’s recommendation at the Department of Laboratory Medicine, Faculty of Medicine, University of Debrecen, Hungary. Non-HDL-C was calculated by the subtraction of total cholesterol and HDL-C. Five aliquots of sera in Eppendorf tubes were stored at −80 °C for subsequent determinations.

### 4.4. Determination of IGFBP-1, IGFBP-3, and IGFBP-rp1 Levels

Serum IGFBP-1, IGFBP-3 and IGFBP-rp1 concentrations were measured using commercially available duoset enzyme linked immunoassays (ELISA, R&D Systems, Abington, UK; Cat. No. DY851, DY675 and DY1334, respectively). Based on the preliminary laboratory experiments, sera were used in 40-fold (IGFBP-1), 1000-fold (IGFBP-3), and 500-fold (IGFBP-rp1) dilutions, respectively. Data were measured in pg/mL in the case of IGFBP-1 and ng/mL in the case of IGFBP-3 and IGFBP-rp1, respectively.

### 4.5. Measurement of sICAM-1 and sVCAM-1 Levels

Serum soluble intercellular adhesion molecule-1 (sICAM-1) and soluble vascular cell adhesion molecule-1 (sVCAM-1) were determined using the ELISA method (R&D Systems, Abington, UK; Cat. No. DCD540 and DVC00, respectively).

### 4.6. Statistical Methods

Statistical analyses were performed using Statistica 13.5.0.17 software (TIBCO Software Inc., San Ramon, CA, USA). Graphs were made using GraphPad Prism 8.01 (GraphPad Prism Software Inc., Boston, MA, USA).

Before the study, analysis of statistical power was conducted with SPH Analytics online calculator (SPH Analytics LTD., Alpharetta, GA, USA) to determine the expected changes in serum IGFBPs during the one-year follow-up with the following estimated serum IGFBP levels: 5000 (SD = 4500) vs. 10,000 (SD = 4500) pg/mL (IGFBP-1); 1000 (SD = 400) vs. 500 (SD = 200) ng/mL (IGFBP-3); 150 (SD = 25) vs. 120 (SD = 25) ng/mL (IGFBP-rp1), respectively. The required sample size was n = 17 for IGFBP-1, n = 10 for IGFBP-3, and n = 15 for IGFBP-rp1, respectively. The alpha level was 0.05 with 0.8 (80%) of desired power. The estimated serum IGFBPs were calculated based on data from the literature and preliminary laboratory experiments.

Normality of variables was checked using the Kormogorov–Smirnov test. Continuous variables were expressed as median (interquartile ranges) or mean (standard deviation). Group comparisons were conducted using the Kruskal–Wallis H test (with Bonferroni correction) for continuous variables. A repeated-measures ANOVA (with Bonferroni correction) was employed to analyze the changes in continuous variables during the 26- and 52-week semaglutide or sitagliptin treatment compared to baseline. Pearson’s correlation was used to determine the strength and direction of the linear relationship between the two continuous variables. Backward stepwise multiple regressions were designed to determine significant predictor(s) of IGFBP-1 and IGFBP-3. Probability values ≤0.05 were considered statistically significant.

## 5. Conclusions

This is the first clinical study to evaluate changes in the levels of three key IGFBPs during long-term treatment with semaglutide and sitagliptin. Our results indicate that both agents are capable of increasing IGFBP-1 levels, highlighting the clinical relevance of incretin-based antidiabetic therapies in routine practice and suggesting a potential role of the GH/IGF-1 axis and IGFBP expression in the management of T2DM and in the prevention of its complications. However, since the exact normal ranges of IGFBP levels are not known, further large-scale multicenter studies specifically designed to assess their impact on diabetes-related complications are necessary. Measurement of IGFBP levels may contribute to a deeper understanding of the mechanisms of action of antidiabetic agents, potentially enabling the identification of novel therapeutic targets and agents.

## Figures and Tables

**Figure 1 ijms-26-10404-f001:**
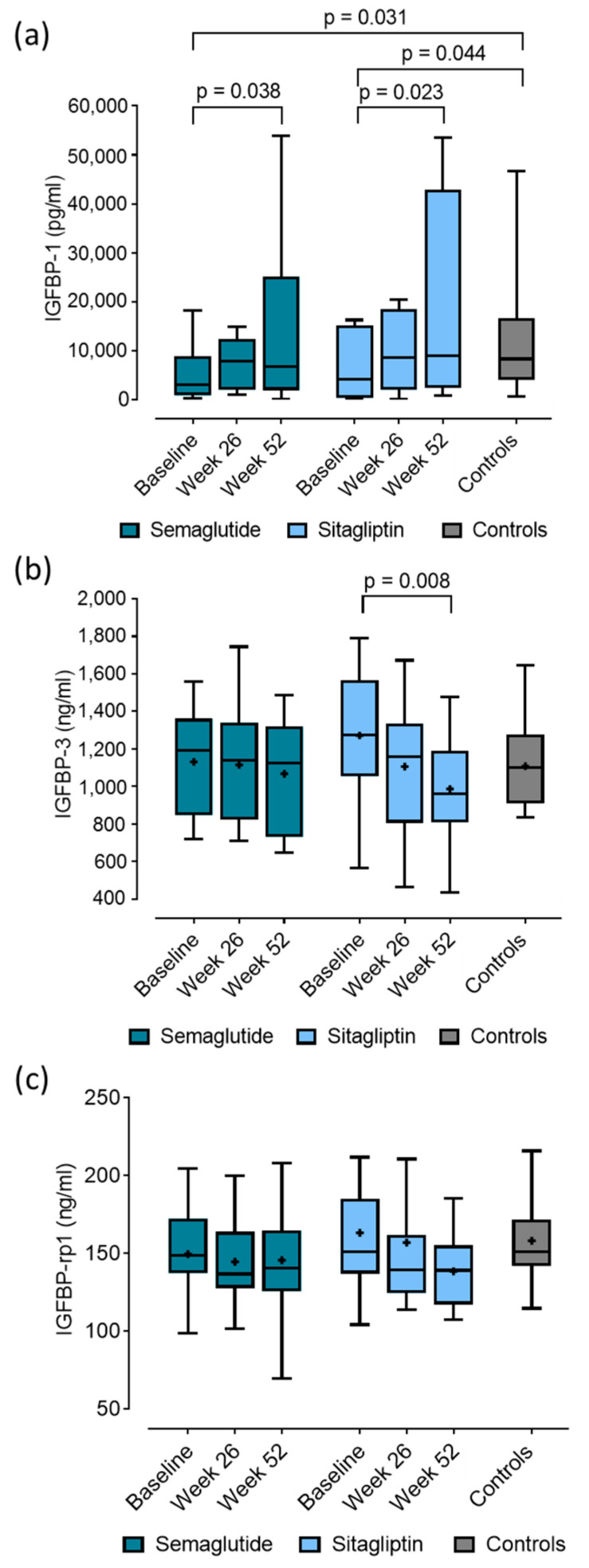
Baseline and follow-up serum concentrations of (**a**) insulin like growth-factor 1 binding protein 1 (IGFBP-1); (**b**) insulin like growth-factor 1 binding protein 3 (IGFBP-3); and (**c**) insulin like growth-factor 1 binding protein related protein 1 (IGFBP-rp1) in patients with type 2 diabetes during 26- and 52-week semaglutide and sitagliptin treatments and age-, sex-, and body mass index- matched controls. Solid lines, boxes and whiskers represent medians, lower/upper quartiles and minimum/maximum values, respectively. + indicates means.

**Figure 2 ijms-26-10404-f002:**
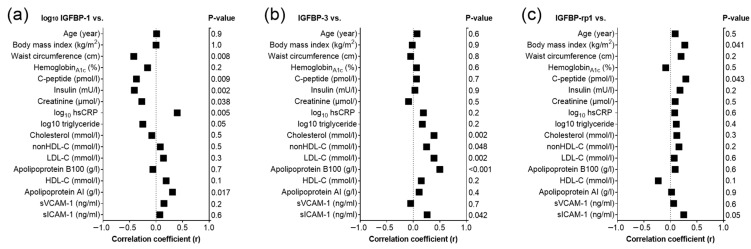
Correlations between main anthropometric, laboratory parameters and (**a**) insulin like growth-factor 1 binding protein 1 (IGFBP-1); (**b**) insulin like growth-factor 1 binding protein 3 (IGFBP-3); and (**c**) insulin like growth-factor 1 binding protein related protein 1 (IGFBP-rp1) in the overall study population. Abbreviations: hsCRP, high-sensitivity C-reactive protein; HDL-C, high-density lipoprotein cholesterol; LDL-C, low-density lipoprotein cholesterol; sICAM-1, soluble intercellular adhesion molecule-1; sVCAM-1, soluble vascular cell adhesion molecule-1.

**Table 1 ijms-26-10404-t001:** Baseline anthropometric and laboratory parameters of T2DM patients and controls.

	Patients with T2DM		Controls
	Semaglutide Group (Baseline)	Sitagliptin Group (Baseline)	
Number of patients	18 (12 m/6 f)	16 (7 m/9 f)	31 (10 m/21 f)
Age (year)	57.9 ± 10.6	59.4 ± 11.9	55.2 ± 5.0
Body mass index (kg/m^2^)	37.96 ± 10.64	31.26 ± 2.75	37.89 ± 9.64
Waist circumference (cm)	126.4 ± 21.8	126.6 ± 25.1	114.8 ± 21.0
Fasting glucose (mmol/L)	8.1 (7.10–11.80) #	8.90 (7.60–10.80) §	5.35 (4.95–6.05)
Fructosamine (mmol/L)	322.39 ± 87.95 #	299.00 ± 68.80 §	239.35 ± 55.37
Hemoglobin_A1c_ (%)	8.1 ± 1.7 #	8.1 ± 1.3 §	6.0 ± 1.3
Insulin (mU/L)	19.4 (15.2–27.5)	26.2 (20.5–41.6) §	13.7 (8.9–19.8)
C-peptide (pmol/L)	1370 (1270–1800) #	1430 (1140–2730) §	826.5 (502–976)
Creatinine (µmol/L)	78.50 ± 15.07	71.47 ± 10.18	68.41 ± 15.21
eGFR (mL/min/1.73 m^2^)	87 (74–90)	90 (73–90)	90 (89–90)
hsCRP (mg/L)	2.3 (1.7–5.8) #	5.5 (2.9–13.8)	11.6 (3.0–18.0)
AST (U/L)	21 (16–25)	27 (21–35)	20.5 (17–24)
ALT (U/L)	24 (17–37)	34 (23–46)	25 (16–37)
GGT (U/L)	33 (24–49)	38 (22–87)	27 (18–39)
Triglyceride (mmol/L)	1.72 (1.50–3.11)	2.00 (1.30–3.40)	1.70 (1.0–2.28)
Total cholesterol (mmol/L)	5.49 ± 1.36	5.85 ± 2.24	5.14 ± 0.99
HDL-C (mmol/L)	1.27 ± 0.30	1.34 ± 0.52	1.30 ± 0.32
Apolipoprotein AI (g/L)	1.55 ± 0.24	1.62 ± 0.33	1.59 ± 0.27
non-HDL-C (mmol/L)	3.98 ± 0.95	4.09 ± 1.95	3.58 ± 1.29
LDL-C (mmol/L)	3.13 ± 0.88	3.28 ± 1.51	3.28 ± 0.84
Apolipoprotein B100 (g/L)	1.03 ± 0.28	1.26 ± 0.50	1.04 ± 0.25
sICAM-1 (ng/mL)	262.1 ± 65.4	277.6 ± 63.8	280.4 ± 51.7
sVCAM-1 (ng/mL)	783.9 ± 212.2	771.1 ± 181.9	640.4 ± 169.1

Notes: eGFR, estimated glomerular filtration rate; AST, aspartate transaminase, ALT, alanine transaminase; GGT, gamma-glutamyl transferase; HDL-C, high-density lipoprotein cholesterol; hsCRP, high-sensitivity C-reactive protein; LDL-C, low-density lipoprotein cholesterol; sICAM-1, soluble intercellular adhesion molecule-1; sVCAM-1, soluble vascular cell adhesion molecule-1; T2DM, type 2 diabetes mellitus. # marks *p* < 0.05 between semaglutide and the control group; § marks *p* < 0.05 between sitagliptin and the control group. Statistical differences were calculated using a Kruskal–Wallis ANOVA.

**Table 2 ijms-26-10404-t002:** Changes in laboratory parameters in patients with T2DM after 52-week semaglutide treatment.

	Baseline	Week 26	Week 52
Number of patients	18 (12 m/6 f)		
Age (year)	57.9 ± 10.6		
Body mass index (kg/m^2^)	37.96 ± 10.64	35.28 ± 9.43 †	34.88 ± 10.22 §
Waist circumference (cm)	126.4 ± 21.8	119.7 ± 21.6 †	115.5 ± 20.9 §
Fasting glucose (mmol/L)	8.1 (7.1–11.8)	7.5 (5.0–8.6) †	7.7 (5.3–10.4) §
Fructosamine (mmol/L)	322.4 ± 87.9	260.4 ± 39.2 †	251.5 ± 37.7 §
Hemoglobin_A1c_ (%)	8.08 ± 1.65	6.86 ± 1.12 †	6.57 ± 0.95 §
Insulin (mU/L)	19.4 (15.2–27.5)	21.6 (12.3–28.1)	19.3 (9.1–30.2)
C-peptide (pmol/L)	1370 (1270–1800)	1340 (733–2110)	1240 (1040–1610)
Creatinine (µmol/L)	78.50 ± 15.07	77.94 ± 30.64	71.56 ± 16.67
eGFR (mL/min/1.73 m^2^)	86.5 (74–90)	90 (78–90)	90 (84–90)
hsCRP (mg/L)	2.3 (1.7–5.8)	2.30 (1.30–6.26)	2.10 (1.10–2.40)
AST (U/L)	21 (17–25)	20.5 (14–28)	17 (15–26)
ALT (U/L)	27 (21–37)	26 (18–32)	21 (16–31)
GGT (U/L)	34 (27–56)	33 (23–45)	35 (22–50)
Triglyceride (mmol/L)	1.72 (1.50–3.11)	1.575 (1.00–2.17)	1.585 (1.0–2.50)
Total cholesterol (mmol/L)	5.49 ± 1.36	4.83 ± 1.31	4.79 ± 1.00
HDL-C (mmol/L)	1.27 ± 0.30	1.40 ± 0.37 †	1.43 ± 0.38 §
Apolipoprotein AI (g/L)	1.55 ± 0.24	1.49 ± 0.23	1.54 ± 0.27
non-HDL-C (mmol/L)	3.98 ± 0.95	3.34 ± 1.08	3.35 ± 0.82 §
LDL-C (mmol/L)	3.13 ± 0.88	2.85 ± 1.06	2.72 ± 0.84 §
Apolipoprotein B100 (g/L)	1.03 ± 0.28	0.96 ± 0.31	0.99 ± 0.25
sICAM-1 (ng/mL)	262.1 ± 65.4	238.9 ± 59.7	250.8 ± 49.7
sVCAM-1 (ng/mL)	783.9 ± 212.2	763.9 ± 162.1	748.3 ± 150.1

Notes: eGFR, estimated glomerular filtration rate; AST, aspartate transaminase, ALT, alanine transaminase; GGT, gamma-glutamyl transferase; HDL-C, high-density lipoprotein cholesterol; hsCRP, high-sensitivity C-reactive protein; LDL-C, low-density lipoprotein cholesterol; sICAM-1, soluble intercellular adhesion molecule-1; sVCAM-1, soluble vascular cell adhesion molecule-1. † marks *p* < 0.05 between baseline and week 26; § marks *p* < 0.05 between baseline and week 52. Statistical differences were calculated using a repeated-measures ANOVA with Bonferroni correction.

**Table 3 ijms-26-10404-t003:** Changes in laboratory parameters in patients with T2DM after a 52-week sitagliptin treatment.

	Baseline	Week 26	Week 52
Number of patients	16 (7 m/9 f)		
Age (year)	59.4 ± 11.9		
Body mass index (kg/m^2^)	31.26 ± 2.75	31.10 ± 2.98	30.911 ± 2.9 §
Waist circumference (cm)	126.60 ± 25.08	125.25 ± 29.18	124.25 ± 29.58
Fasting glucose (mmol/L)	8.90 (7.60–10.80)	7.50 (5–7.9) †	8.00 (5.7–9.3)
Fructosamine (mmol/L)	299.0 ± 68.8	262.6 ± 51.2	253.4 ± 28.7 §
Hemoglobin_A1c_ (%)	8.13 ± 1.29	6.91 ± 0.95 †	7.11 ± 1.07 §
Insulin (mU/L)	26.2 (20.5–41.6)	26.0 (18.7–40.2)	21.95 (12.5–26.5)
C-peptide (pmol/L)	1430 (1140–2730)	1870 (966–2020)	1110 (904.5–1675)
Creatinine (µmol/L)	71.47 ± 10.18	70.571 ± 15.301	71.20 ± 14.26
eGFR (mL/min/1.73 m^2^)	90 (73–90)	90 (85–90)	90 (85–90)
hsCRP (mg/L)	5.5 (2.9–13.8)	4.7 (2.3–11.7)	8.7 (2.4–14.7)
AST (U/L)	27 (21–35)	23 (18–32)	28.5 (16–36)
ALT (U/L)	34 (23–46)	29.5 (21–43)	36 (25–46)
GGT (U/L)	38 (22–87)	39 (20–52)	39 (25–44)
Triglyceride (mmol/L)	2.00 (1.30–3.40)	2.08 (1.55–2.9)	1.85 (1.3–3.1)
Total cholesterol (mmol/L)	5.85 ± 2.24	5.264 ± 1.32	5.31 ± 1.57
HDL-C (mmol/L)	1.34 ± 0.52	1.303 ± 0.44	1.338 ± 0.34
Apolipoprotein AI (g/L)	1.62 ± 0.33	1.53 ± 0.39 †	1.48 ± 0.34 §
non-HDL-C (mmol/L)	4.09 ± 1.95	3.962 ± 1.34	4.00 ± 1.52
LDL-C (mmol/L)	3.28 ± 1.51	3.09 ± 1.08	3.21 ± 1.2
Apolipoprotein B100 (g/L)	1.26 ± 0.50	1.10 ± 0.4	1.18 ± 0.41
sICAM-1 (ng/mL)	277.6 ± 63.8	291.9 ± 64.5	284.0 ± 43.4
sVCAM-1 (ng/mL)	771.1 ± 181.9	860.2 ± 547.1	691.9 ± 75.0

Notes: eGFR, estimated glomerular filtration rate; AST, aspartate transaminase, ALT, alanine transaminase; GGT, gamma-glutamyl transferase; HDL-C, high-density lipoprotein cholesterol; hsCRP, high-sensitivity C-reactive protein; LDL-C, low-density lipoprotein cholesterol; sICAM-1, soluble intercellular adhesion molecule-1; sVCAM-1, soluble vascular cell adhesion molecule-1. † marks *p* < 0.05 between baseline and week 26; § marks *p* < 0.05 between baseline and week 52. Statistical differences were calculated using a repeated-measures ANOVA with Bonferroni correction.

## Data Availability

The original contributions presented in this study are included in the article/Appendix A. Further inquiries can be directed to the corresponding author.

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
