# Peer review of "Long-Term Effects of Semaglutide and Sitagliptin on Circulating IGFBP-1, IGFBP-3 and IGFBP-rp1: Results from a One-Year Study in Type 2 Diabetes"

_ijms, 2025, doi:10.3390/ijms262110404_

Round 1
Reviewer 1 Report
Comments and Suggestions for Authors
The current manuscript, reported by Harangi et al., is interesting, as it shows changes in IGFBPs in patients with type 2 diabetes who are using a GLP-1 receptor agonist (semaglutide) and a DPP-4 inhibitor (sitagliptin). The study is well-designed and well-written
Minor issue
lines 256-257, the authors should extend their explanation regarding the unexpected results of IGFBP-3 among patients using sitagliptin and semaglutide. Consider adding the broad effect of DPP-4 inhibitors on both GLP-1 and GIP, thus restoring the incretin balance, while GLP-1 Rc agonists with supraphysiologic doses provide GLP-1 signaling only. Please add a reference regarding this issue.
Reviewer 2 Report
Comments and Suggestions for Authors
This study investigates the effects of semaglutide and sitagliptin, added to metformin, on serum IGFBP-1, IGFBP-3, and IGFBP-rp1 levels in overweight and obese patients with T2DM at weeks 26 and 52. The topic is of interest and as far as I am aware, and as the authors also state, no previous studies have directly examined these biomarkers in diabetic patients treated with semaglutide or sitagliptin, which makes this research potentially novel and relevant, paving the basis for further investigation.
However, in its current form, the manuscript has several MAJOR methodological limitations that must be addressed before it can be considered for publication:
- The study is not registered in a public trial registry (e.g., ClinicalTrials.gov), and no explanation is provided.
- The “Materials and Methods section” does not describe the randomization process, including the criteria and method by which participants were assigned to groups (semaglutide vs sitagliptin) (e.g., computer-generated random sequence? Or?). Clear details are necessary to assess the validity of the study design.
- The method of sample size determination (power calculation, ... α, β) must be specified for randomized trails/studies ( see CONSORT requirements). With such small groups (18 vs. 16 patients), the reliability of the findings may be questioned unless this is justified or at least acknowledged as a limitation.
- No adverse events related to the two drugs used are reported in the manuscript.
- Data on any diet followed by patients and physical activity (which might influence the results) are missing. These are only vaguely mentioned in the study limitations.
